# Serum Antibody Activity against Poly-*N*-Acetyl Glucosamine (PNAG), but Not PNAG Vaccination Status, Is Associated with Protecting Newborn Foals against Intrabronchial Infection with *Rhodococcus equi*

Noah D. Cohen,[a] Susanne K. Kahn,[a] Colette Cywes-Bentley,[b] Sophia Ramirez-Cortez,[a] Amanda E. Schuckert,[a] Mariana Vinacur,[b] Angela I. Bordin,[a] Gerald B. Pier[b]

[a]Equine Infectious Disease Laboratory, Department of Large Animal Clinical Sciences, College of Veterinary Medicine & Biomedical Sciences, Texas A&M University, College Station, Texas, USA
[b]Department of Medicine, Brigham & Women's Hospital, Harvard Medical School, Boston, Massachusetts, USA

**ABSTRACT** *Rhodococcus equi* is a prevalent cause of pneumonia in foals worldwide. Our laboratory has demonstrated that vaccination against the surface polysaccharide $\beta$-1→6-poly-*N*-acetylglucosamine (PNAG) protects foals against intrabronchial infection with *R. equi* when challenged at age 28 days. However, it is important that the efficacy of this vaccine be evaluated in foals when they are infected at an earlier age, because foals are naturally exposed to virulent *R. equi* in their environment from birth and because susceptibility is inversely related to age in foals. Using a randomized, blind experimental design, we evaluated whether maternal vaccination against PNAG protected foals against intrabronchial infection with *R. equi* 6 days after birth. Vaccination of mares *per se* did not significantly reduce the incidence of pneumonia in foals; however, activities of antibody against PNAG or for deposition of complement component 1q onto PNAG was significantly ($P < 0.05$) higher among foals that did not develop pneumonia than among foals that developed pneumonia. Results differed between years, with evidence of protection during 2018 but not 2020. In the absence of a licensed vaccine, further evaluation of the PNAG vaccine is warranted, including efforts to optimize the formulation and dose of this vaccine.

**IMPORTANCE** Pneumonia caused by *R. equi* is an important cause of disease and death in foals worldwide for which a licensed vaccine is lacking. Foals are exposed to *R. equi* in their environment from birth, and they appear to be infected soon after parturition at an age when innate and adaptive immune responses are diminished. Results of this study indicate that higher activity of antibodies recognizing PNAG was associated with protection against *R. equi* pneumonia, indicating the need for further optimization of maternal vaccination against PNAG to protect foals against *R. equi* pneumonia.

**KEYWORDS** foal, poly-*N*-acetyl glucosamine, *Rhodococcus equi*, antibody, pneumonia, veterinary vaccine development

Infections, notably pneumonia, are leading causes of disease and death in foals (1, 2), and *Rhodococcus equi* is the most common cause of severe pneumonia in foals (3–5). Cumulative incidence and case-fatality proportions for *R. equi* pneumonia in foals can be high, and treatment is prolonged, expensive, associated with adverse effects, and not uniformly successful (3, 4). Thus, *R. equi* pneumonia is a disease of importance to the equine breeding industry worldwide (4, 5).

Efforts to control and prevent *R. equi* foal pneumonia have had limited effectiveness. Screening methods for early detection are expensive, labor-intensive, and imperfect (6–9).

Address correspondence to Noah D. Cohen, ncohen@cvm.tamu.edu, or Gerald B. Pier, gbpier@bwh.harvard.edu.

The disease primarily occurs at large, well-managed farms (10, 11), indicating that neither neglect nor poor hygiene is a determinant of infection. Chemoprophylaxis with macrolides is not an acceptable approach because of concerns about increasing antimicrobial resistance in *R. equi* strains (12, 13) (and possibly other bacteria) and because evidence of effectiveness has been conflicting (14, 15). Transfusion of plasma for prevention can be partially effective, but it is expensive and labor-intensive relative to vaccination (7, 16–18). Although there is not a licensed vaccine against *R. equi*, our laboratories have demonstrated that vaccinating mares with deacetylated $\beta$-1→6 poly-*N*-acetyl-glucosamine (PNAG) raises antibodies against the native acetylated PNAG antigen expressed on the *R. equi* surface that can protect foals against intrabronchial infection with virulent *R. equi* administered at 28 days of age (19). Along with the antibodies targeting PNAG, complement and neutrophils and lymphocytes armed with antibody to PNAG contribute to killing of *R. equi*, protecting foals against pneumonia (19–21).

Epidemiological and clinical observations indicate that foals are exposed to virulent *R. equi* from birth and predominately infected when very young and more susceptible to infection (14, 22–24). Evidence exists that crucial aspects of immune responses differ between newborn foals and older foals that are pertinent to anti-PNAG-mediated protection. Complement activity is reportedly lower among foals during the first week after foaling than among foals that are 4 weeks of age (25–32), and these lower concentrations of complement factors are associated with decreased killing of *R. equi* by neutrophils (27, 28, 30, 32). Neutrophils contribute to protection against *R. equi* (33–35), and functional responses of neutrophils of newborn foals are lower than those of 4-week-old foals (36–39). Given that complement and neutrophils work in concert with anti-PNAG antibodies to kill *R. equi*, it is important to determine whether the protection we have observed in 4-week-old foals (19) can be replicated in foals that are <1 week old. On the other hand, maternal vaccination might contribute to cell-mediated immune responses independent of complement or neutrophil activity: peripheral blood mononuclear cells from foals of vaccinated mares produced significantly more interferon gamma in response to stimulation with *R. equi* than did those of foals from unvaccinated mares, and interferon gamma production was significantly reduced by treatment with dispersin B, which hydrolyzes PNAG (19). We therefore investigated whether vaccinating mares against PNAG protected their foals against experimental infection with *R. equi* 6 days after birth.

## RESULTS

**Antibody activities in mare and foal serum.** Prior to vaccination, antibody activities to PNAG were low among study mares and there was no significant difference ($P = 0.6269$) between the vaccinated mares ($n = 14$) and control mares ($n = 8$) (see Fig. S1 in the supplemental material); however, irrespective of vaccine group, antibody activities were significantly ($P = 0.0208$) higher for mares in 2020 than in 2018. Antibody activities against PNAG were significantly higher in serum from vaccinated mares and their foals than in the corresponding samples from controls (Table 1; Fig. 1 and 2). There was no significant effect of year ($P = 0.7120$) or year by group interaction ($P = 0.3530$) for PNAG antibody activities of mares, nor were effects of year ($P = 0.5530$) or year by group interaction (0.7034) significant for antibody activities of foals. C'1q deposition activity, determined as readings of optical density at 405 nm ($OD_{405}$), were significantly higher in sera from vaccinated mares and their foals than the corresponding samples from controls (Table 1; Fig. 3 and 4). There was no significant effect of year ($P = 0.6503$) or year by group interaction ($P = 0.95554$) for C'1q deposition activity of mares, nor were effects of year ($P = 0.5514$) or year by group interaction (0.3903) significant for C'1q deposition activity of foals.

PNAG antibody activities and C'1q deposition activity ($\log_{10}$ transformed) from mares (Fig. 5) and foals (Fig. 6) were significantly correlated ($P = 0.0035$ and $P = 0.0004$, respectively). Antibody activities against PNAG and C'1q deposition activity in foals at

**TABLE 1** Antibody activities for PNAG and C′1q OD$_{405}$ readings at age 1 day (approximately 24 h) among mares and foals in the vaccinated group (14 mares and their foals) and the unvaccinated control group (8 mares and their foals)

| Antigen | Mean value (95% CI) in[a]: | | P value |
| | Vaccinees | Controls | |
| --- | --- | --- | --- |
| Mares | | | |
| PNAG | 7,196 (1,506 to 34,390) | 112 (32 to 391) | <0.0001 |
| C′1q deposition | 1.368 (0.579 to 2.156) | 0.132 (−0.497 to 0.761) | 0.0060 |
| Foals | | | |
| PNAG | 3,215 (1,063 to 9,722) | 49 (16 to 147) | <0.0001 |
| C′1q deposition | 1.510 (0.764 to 2.256) | 0.136 (−0.460 to 0.731) | 0.0018 |

[a]Values represent mean estimates (CI, confidence interval) derived from generalized linear modeling with back-transformation of log$_{10}$-transformed data. Values of PNAG are endpoint antibody activities calculated by linear regression using a final OD$_{405}$ value of 0.5 to determine the reciprocal of the maximal serum dilution reaching this value. Values for C′1q deposition were determined as the OD$_{405}$ value obtained in the highest serum concentration.

the age of 1 day were strongly ($R = 0.89$ and 0.92, respectively) as well as significantly ($P < 0.0001$ for both) correlated with those of mares (Fig. S2 and S3).

**Challenge dose.** The targeted number of CFU for challenging foals was 5,000 CFU, split evenly between the right and left lungs. The median challenge dose for the 22 foals was 7,200 (range, 4,600 to 15,100). The challenge dose did not differ significantly ($P = 0.2891$) between vaccinees (median, 7,900; range, 4,600 to 15,100) and controls (median, 6,400; range, 5,100 to 8,400). There was no significant difference ($P = 0.2633$) in the challenge doses administered to foals in year 1 (median, 6,400; range, 4,600 to 8,400) and year 2 (median, 7,900; range, 4,900 to 15,100). In year 1, the challenge dose did not differ significantly ($P = 0.7758$) between vaccinated foals (median, 6,800; range, 4,600 to 8,400) and unvaccinated foals (median, 6,200; range, 5,400 to 8,400). Similarly, in year 2, the challenge dose did not differ significantly ($P = 0.2547$) between vaccinated foals (median, 8,500; range, 4,900 to 15,100) and unvaccinated foals (median, 7,000; range, 5,100 to 7,900). The CFU with which foals were challenged was not significantly different ($P = 0.7396$) between foals that developed pneumonia (median, 7,200; range, 4,900 to 15,100; $n = 16$ foals) and foals that did not develop pneumonia (median, 7,299; range, 4,600 to 8,400; $n = 6$ foals).

**Clinical outcomes.** The primary clinical outcome was development of pneumonia attributed to *R. equi* infection. The proportion of control foals that developed pneumonia (88%; 7/8) was not significantly ($P = 0.3512$) greater than that for vaccinees (64%; 9/14).

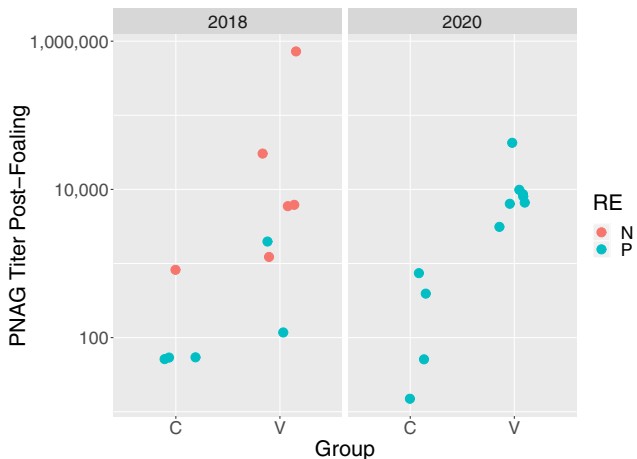

**FIG 1** Antibody activities against PNAG in sera from mares obtained approximately 24 h after foaling, faceted by year (2018 or 2020). Group represents vaccination status (C, control; V, vaccinated against PNAG). Teal circles represent foals that developed pneumonia (P), and coral circles represent foals that remained healthy (N).

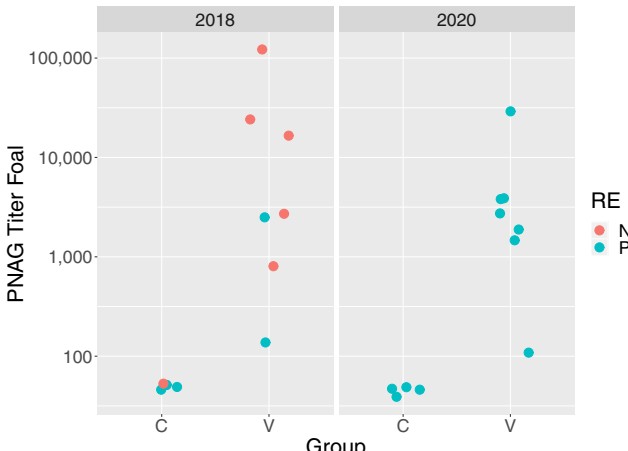

**FIG 2** Antibody activities against PNAG in sera from foals obtained at age approximately 24 h, faceted by year. Group represents vaccination status (C, control; V, vaccinated against PNAG). Teal circles represent foals that developed pneumonia, and coral circles represent foals that remained healthy.

The proportion of foals that developed pneumonia was significantly ($P = 0.0124$) greater in 2020 (100%; 11/11) than in 2018 (45%; 5/11). Among foals that developed pneumonia, the age at onset of pneumonia did not differ significantly between vaccinees (median, 25 days; range, 21 to 32 days) and controls (median, 27 days; range, 23 to 33 days). There were no significant differences between the vaccinated foals and the control foals for any of the findings of twice-daily physical examination, weekly thoracic ultrasonography, weekly hematology, or duration of treatment for pneumonia (Table 2). Using generalized linear modeling, foals with pneumonia had significantly ($P < 0.05$) lower antibody activities for PNAG and C′1q deposition activity than foals that remained healthy, irrespective of vaccination status (Table 3; Fig. 1 and 4).

## DISCUSSION

Mares that were vaccinated against PNAG developed significantly higher serum concentrations of antibodies against PNAG that also deposited C′1q onto PNAG, and these antibodies were transferred to their foals (Fig. 1 to 4; Fig. S1). The concentrations of antibodies recognizing PNAG that also deposited C′1q onto PNAG were strongly correlated (Fig. 5 and 6), and the antibodies in serum of mares to PNAG were strongly correlated with those of their foals (Fig. S2 and S3). Interestingly, some unvaccinated

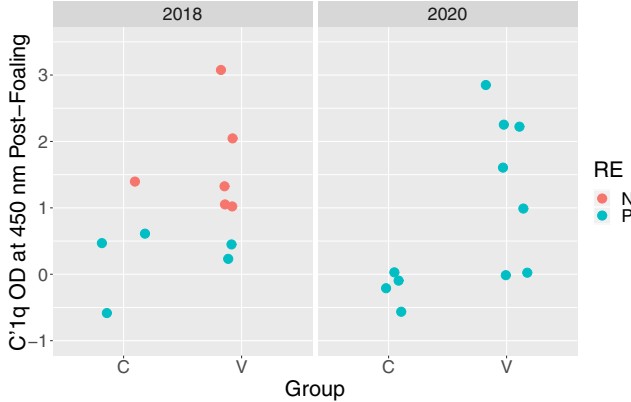

**FIG 3** OD$_{405}$ readings for deposition of C′1q onto PNAG in sera from mares obtained approximately 24 h postfoaling, faceted by year. Group represents vaccination status (C, control; V, vaccinated against PNAG). Teal circles represent foals that developed pneumonia, and coral circles represent foals that remained healthy.

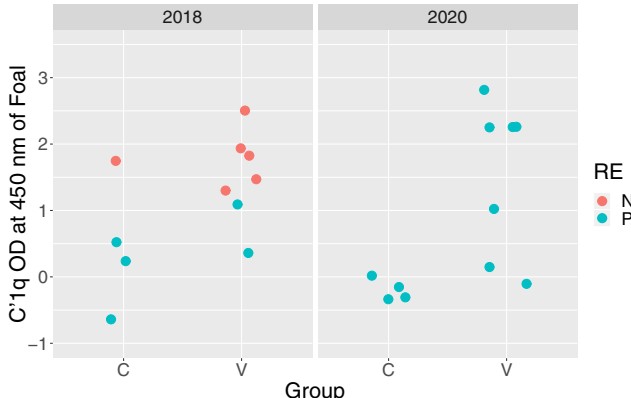

**FIG 4** $OD_{405}$ readings for deposition of C'1q onto PNAG in sera from foals obtained at age approximately 24 h, faceted by year. Group represents vaccination status (C, control; V, vaccinated against PNAG). Teal circles represent foals that developed pneumonia, and coral circles represent foals that remained healthy.

mares and their foals had relatively high concentrations of antibodies that recognized PNAG and that deposited C'1q onto PNAG. We have observed this in field studies (our unpublished data), among unvaccinated mares and their foals.

Although vaccination of mares against PNAG protected their 28-day-old foals against intrabronchial infection with virulent *R. equi* (19), we show here that vaccination of mares failed to protect foals against intrabronchial infection with *R. equi* at age 6 days. One explanation for the difference in results between our previous findings and the results of this study is that younger foals are more susceptible to infection with *R. equi* (23). Reduced activity of complement and functional responses of neutrophils have been reported for foals <7 days of age relative to older foals (36–39). Because complement and neutrophils are essential for killing of *R. equi* mediated by antibodies to PNAG (19), it is plausible that failure of maternal vaccination to protect foals against *R. equi* is attributable to diminished contributions to bacterial killing from complement and neutrophils in younger foals. Alternatively, it is possible that 6-day-old foals have diminished binding to their Fc receptors of antibodies that bind PNAG, resulting in less effective antibody-dependent cellular

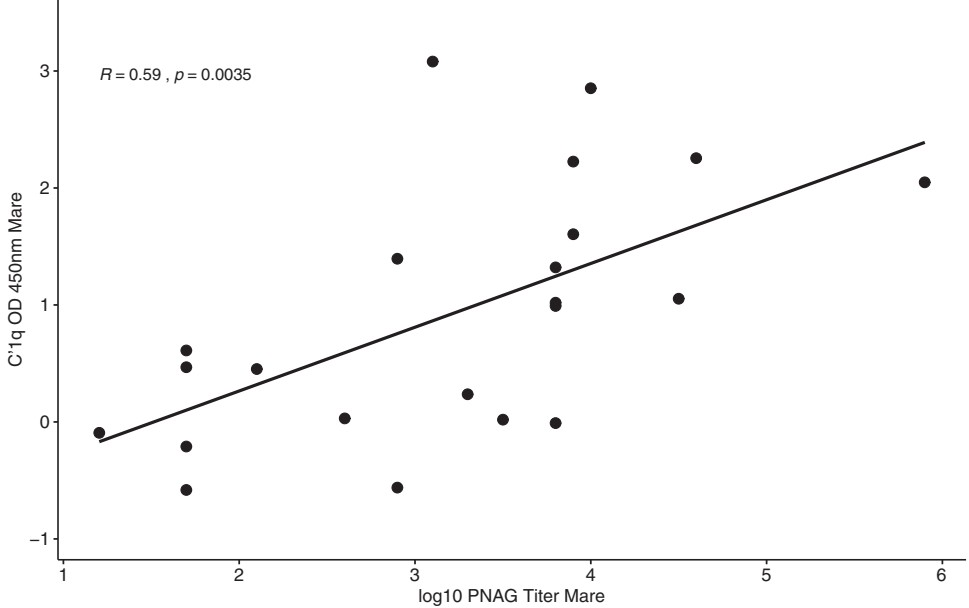

**FIG 5** Antibody activities of antibody binding to PNAG ($\log_{10}$ transformed) and C'1q deposition activity measured in mare sera approximately 24 h postfoaling were significantly ($P = 0.0035$) correlated ($R = 0.59$).

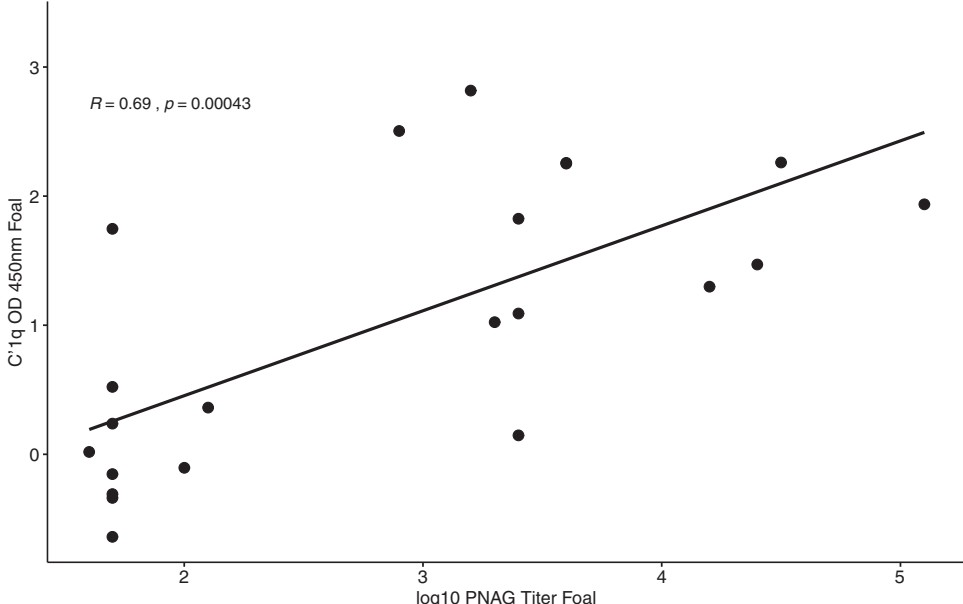

$R = 0.69$ , $p = 0.00043$

**FIG 6** Antibody activities of antibody binding to PNAG ($\log_{10}$ transformed) and C'1q deposition activity measured in foal sera approximately 24 h postfoaling were significantly ($P = 0.0004$) correlated ($R = 0.69$).

cytotoxicity. Although age-related changes in Fc receptor expression have not been reported in foals, expression of Fc receptors for some immunoglobulin isotypes/subisotypes increases with age in mice (40).

Serum anti-PNAG antibody and C'1q deposition among vaccinated mares varied, and in some mares the values were similar to values observed for most unvaccinated mares (Fig. 1 and 3); conversely, 1 unvaccinated mare had relatively high antibody activities for PNAG and C'1q, and this mare's foal was protected against challenge. Indeed, concentrations of antibodies to PNAG and C'1q in both mare and foal sera were significantly ($P < 0.05$) associated with protection against *R. equi* pneumonia (Table 3). Although we cannot exclude the possibility that a mare and her foal were misidentified, careful review of our records did not reveal evidence that this occurred.

**TABLE 2** Values for continuous clinical parameters and results of Wilcoxon rank-sum test comparing foals of vaccinated mares ($n = 14$) with control foals ($n = 8$)

| Clinical or hematological parameter | Median (range) for: | | P value[a] |
| --- | --- | --- | --- |
| | Vaccinees | Controls | |
| Frequency[b] of: | | | |
| Tachypnea[c] | 51.5 (36 to 108) | 60.5 (37 to 80) | 0.1825 |
| Temp >103°F | 6 (0 to 50) | 8.5 (4 to 20) | 0.3362 |
| Temp >102.5°F | 12.5 (1 to 61) | 14 (8 to 28) | 0.5379 |
| Coughing | 5 (0 to 23) | 9.5 (2 to 18) | 0.1074 |
| Frequency (wks) of WBC >13,000 cells/$\mu$l | 2 (0 to 7) | 1.5 (0 to 5) | 0.6959 |
| Maximum WBC (1,000 cells/$\mu$l) | 22.34 (11.67 to 71.74) | 15.72 (12.27 to 32.35) | 0.7135 |
| Frequency (wks) of fibrinogen >400 mg/dl | 3.5 (1 to 9) | 3 (1 to 5) | 0.3848 |
| Maximum fibrinogen (mg/dl) | 800 (400 to 1,200) | 600 (400 to 1,100) | 0.3862 |
| Duration (wks) of ultrasound lesions | 3 (1 to 8) | 3 (2 to 4) | 0.9717 |
| Maximum diam of lesions (mm) | 116.25 (13.0 to 342.0) | 138.0 (32.5 to 212.2) | 0.9203 |
| Sum of diam of lesions (mm) | 201.3 (13.0 to 1,807.4) | 194.3 (72.5 to 549.1) | 0.8154 |
| No. of days treated | | | |
| All foals | 12.5 (0 to 85) | 17 (0 to 24) | 0.4472 |
| Pneumonic foals | 15 (11 to 85) | 19 (8 to 24) | 0.9152 |

[a]Derived by Wilcoxon rank-sum test.
[b]Frequency of observation was twice daily; 2 observations = 1 day.
[c]Defined as a respiratory rate of >40 breaths/min.

**TABLE 3** Antibody activities for PNAG antibody binding and C'1q deposition activity at age 1 day (approximately 24 h) from mares and their foals by whether the foal developed pneumonia (*n* = 16) or remained healthy (*n* = 6) in the vaccinated group (14 mares and their foals) and the unvaccinated control group (8 mares and their foals)

| Antigen | Mean value (95% CI) in[a]: | | *P* value |
|---|---|---|---|
| | Healthy | Pneumonia | |
| Mares | | | |
| PNAG | 10,000 (1,353 to 73,905) | 794 (76 to 8,291) | 0.0470 |
| C'1q deposition | 1.653 (0.86 to 2.45) | 0.643 (−0.292 to 1.576) | 0.0466 |
| Foals | | | |
| PNAG | 4,135 (610 to 28,037) | 360 (38 to 3,392) | 0.0455 |
| C'1q deposition | 1.797 (1.011 to 2.582) | 0.715 (−0.206 to 1.636) | 0.0322 |

[a]Values represent mean estimates (CI, confidence interval) derived from generalized linear modeling with back-transformation of $\log_{10}$-transformed data.

Thus, vaccine formulations that can achieve adequate levels of functional antibodies recognizing PNAG, whether by vaccination or through natural exposure, can protect young foals against infection with *R. equi*. For example, the proportion of foals that developed pneumonia was significantly ($P = 0.01512$) lower among foals with C'1q $OD_{405}$ readings of $\geq 1.0$ (50%; 6/12) than in foals with C'1q $OD_{405}$ readings of $<1.0$ (100%; 10/10). Thus, further optimization of the PNAG vaccine is warranted to improve both the magnitude and consistency of antibody concentrations in mares and their foals. Some variation, however, must be expected because of individual variation in immune responses to immunizations, gestational age, and foal absorption of colostrally transferred antibodies. Although the difference was not significant, more than 1/3 of foals born to vaccinated mares were protected (versus 17% of controls).

There was a clear difference in results between study years. In 2018, more foals were resistant to *R. equi* infection, and this resistance was correlated with antibody activities. Indeed, no foals with C'1q $OD_{405}$ readings of $>1.1$ developed pneumonia in 2018, whereas in 2020 all foals developed pneumonia. This difference among years was not attributable to differences in the challenge dose or in antibody activities. Although we attempted to keep management practices consistent between years, the location where mares and foals were housed changed between years, as did personnel. Unfortunately, we could not examine the effects of interaction between study year and pneumonia status on antibody activities using multivariable regression because of complete separation (i.e., all foals in 2020 developed pneumonia).

Another explanation for the failure to observe overall significant protection against challenge between vaccinated and control foals is that our challenge model might have been too severe. A prior study conducted in Kentucky used $1 \times 10^3$ CFU of a virulent strain of *R. equi* to infect 4 foals $\leq 7$ days of age, and 2 of these foals developed clinical signs of disease (23). In an attempt to achieve a proportion of $>50\%$ of foals with clinical signs among control foals, we used a higher challenge dose in this study. Thus, our challenge dose might have exceeded the protective benefits of PNAG antibodies among some foals at an age when they are highly susceptibility to infection (23). Bolus experimental infections in fluid do not mirror natural infections occurring via inhalation, which likely contain many fewer CFU and do not include aspiration of fluid. Consequently, findings of our study may not be able to be extrapolated to expected effects under natural conditions.

In summary, our results do not indicate that vaccination of mares with a formulation of PNAG protected foals of $<1$ week of age against an intrabronchial challenge dose of $5 \times 10^3$ CFU of a virulent strain of *R. equi*. However, protection was associated with antibody activity, particularly in the C'1q deposition assay, which is indicative of functional antibody activity. Further refinement of the vaccine and challenge model appear warranted prior to evaluation of this vaccine under field conditions, especially in relation to the diagnosis of *R. equi* pneumonia that is acquired in the first week of life.

## MATERIALS AND METHODS

**Ethics statement.** All procedures for this study were reviewed and approved by the Texas A&M Institutional Animal Care and Use Committee (protocol numbers AUP IACUC 2016-0233 and IACUC 2019-0347) and the University Institutional Biosafety Committee (permit number IBC2017-105). The foals used in this study were university owned, and permission for their use was provided in compliance with the Institutional Animal Care and Use Committee procedures.

**Study population and vaccination of mares.** A group of 22 healthy, pregnant Quarter Horse mares and foals were used for this project. All foals were healthy, had adequate transfer of passive immunity as assessed by a semiquantitative immunoassay (SNAP foal IgG test; IDEXX, Westbrook, ME, USA), and had results of complete blood counts (CBCs) within reference ranges. Mares were randomly assigned during midgestation to 1 of 2 experimental groups: group 1 mares were vaccinated with PNAG ($n = 14$), and group 2 mares were sham vaccinated with 1 ml of sterile medical-grade physiological saline (i.e., 0.9% NaCl) solution (PSS). Mares in the vaccine group received 200 $\mu$g of synthetic pentamers of $\beta$-1→6-linked glucosamine conjugated to tetanus toxoid (ratio of oligosaccharide to protein, 35 to 39:1; 5GlcNH$_2$-TT; AV0328; Alopexx Enterprises, LLC, Concord, MA, USA) diluted to 900 $\mu$l in PSS combined with 100 $\mu$l of a water-in-oil adjuvant (Montanide Gel 01; Seppic, Inc., Courbevoie, France). Mares were vaccinated and given boosters (group 1) or sham vaccinated (group 2) at 6 and 3 weeks prior to their projected foaling dates. This sample size was based on the magnitude of prior observed effects of the vaccine (19) and the following assumptions: (i) significance of 5%; (ii) power of >80%; (iii) 75% (6/8) of control foals developing pneumonia; and (iv) 7% (1/14) of principal foals developing pneumonia. The study was conducted over 2 years: each year, we included 7 vaccinated mares and their foals and 4 unvaccinated mares and their foals.

**Infection and monitoring of foals.** Prior to infection with *R. equi*, each foal's lungs were auscultated and examined by thoracic ultrasonography to document absence of preexisting pulmonary disease. Ultrasound examinations were repeated weekly after infection to monitor for lung lesions. At age 6 days, foals were sedated and infected transendoscopically with 15 ml of *R. equi* suspension infused into each mainstem bronchus (right and left).

Foals were infected with a virulent strain of *R. equi* originally isolated from a pneumonic foal from Texas (strain EIDL 5-331) that we have used previously to infect foals at age 28 days (19, 41, 42). This strain was streaked onto a brain heart infusion (BHI) agar plate (Bacto brain heart infusion; BD, Becton, Dickinson and Company, Sparks, MD, USA). One CFU was incubated overnight at 37°C in 50 ml of BHI broth on an orbital shaker at approximately 240 rpm. The bacterial cells were washed 3 times with $1\times$ phosphate-buffered saline (PBS) by centrifugation for 10 min at $3,000 \times g$ and 4°C. The final washed pellet was resuspended in 30 ml of sterile medical-grade PBS to a final concentration of approximately 167 CFU/ml, yielding a total CFU count of approximately $5 \times 10^3$ in 30 ml. Half of this challenge dose (15 ml with $2.5 \times 10^3$ CFU) was administered transendoscopically to the left mainstem bronchus and the other half (15 ml with $2.5 \times 10^3$ CFU) was administered to the right mainstem bronchus. Approximately 200 $\mu$l of challenge dose was saved to confirm the concentration (dose) administered and to verify virulence of the isolate using multiplex PCR (43). This challenge dose was based on evidence that younger foals are more susceptible to infection (23), being 5-fold greater than that reported to cause pneumonia in 50% of foals infected at 4 to 7 days of age (23) and being 200-fold less than the dose we have used reliably to cause pneumonia in foals infected at approximately 28 days of age (19, 41, 42).

**Sample collections from mares and foals.** Blood samples were drawn into clot tubes from mares and foals between 18 and 24 h after birth to harvest serum. Blood was also collected in EDTA tubes for CBC testing at ages 18 to 24 h, 6 days (prior to infection) and then weekly after infection through the age of 55 days. Transendoscopic tracheobronchial aspiration (T-TBA) was performed at the time of onset of clinical signs for any foals developing pneumonia and at approximately age 55 days for all foals (end of study) by washing the tracheobronchial tree with sterile PBS solution delivered through a triple-lumen, double-guarded sterile tubing system (MILA International, Inc., Erlanger, KY, USA).

**PNAG ELISA.** Humoral antibody responses were assessed in sera by indirectly quantifying concentrations by enzyme-linked immunosorbent assay (ELISA) from absorbance values of PNAG-specific total IgG. ELISA plates (MaxiSorp, Nalge Nunc International, Rochester, NY, USA) were coated with 0.6 $\mu$g/ml of purified PNAG (19) diluted in sensitization buffer (0.04 M PO$_4$, pH 7.2) overnight at 4°C. Plates were washed 3 times with PBS with 0.05% Tween 20, blocked with 120 $\mu$l PBS containing 1% skim milk for 1 h at 37°C, and washed again. Serum samples were added in 100-$\mu$l volumes, in duplicate, to the ELISA plate and incubated for 1 h at 37°C. Samples were initially diluted in incubation buffer (PBS with 1% skim milk and 0.05% Tween 20) to 1:100 and then serially diluted 2-fold to determine total IgG antibody activities. A positive control from a horse previously immunized with the 5GlcNH$_2$-TT vaccine and known to have a high titer and normal horse serum known to have a low titer were included in each assay for total IgG antibody activities. After 1 h incubation at 37°C, the plates were washed 3 times as described above. Rabbit anti-horse IgG whole molecule conjugated to alkaline phosphatase (Sigma-Aldrich, St. Louis, MO, USA) was used to detect binding. Plates were washed again, and PNPP (*p*-nitrophenyl phosphate) substrate (1 mg/ml) was added to plates. Plates were incubated for 15 to 60 min at 22°C in the dark. Optical densities were determined at 405 nm by using microplate readers. Total IgG endpoint antibody activities were calculated by linear regression using a final OD$_{405}$ value of 0.5 to determine the reciprocal of the maximal serum dilution reaching this value.

**C′1q deposition assays.** Quantitation of the serum antibody-mediated deposition of equine complement component C′1q onto purified PNAG was determined by ELISA. ELISA plates were sensitized with PNAG, and then horse serum was added in 50-$\mu$l volumes, after which 50 $\mu$l of 6% or 10% intact, normal horse serum was added. After 60 min incubation at 37°C, plates were washed and either 100 $\mu$l of affinity-purified goat anti-human C′1q, which also binds to equine C′1q, diluted 1:20,000 in incubation buffer was added or the same antibody directly conjugated to alkaline phosphatase was added at

1.5 µg/ml. Plates were incubated at room temperature for 60 min. For assays with unconjugated anti-C´1q, plates were washed, and 100 µl of rabbit anti-goat IgG whole molecule conjugated to alkaline phosphatase diluted 1:1,000 in incubation buffer added; a 1-h incubation at room temperature was then carried out. Washing and development of the color indicator were then carried out as described above, and values for C´1q deposition were determined as the $OD_{405}$ value obtained in the highest serum concentration. This is referred to as the C´1q deposition activity.

**Clinical monitoring.** From birth until age 5 days, foals were observed twice daily for signs of disease. Beginning at age 5 days (the day prior to infection), rectal temperature, heart rate, respiratory rate, signs of increased respiratory effort (abdominal lift, flaring nostrils), presence of abnormal lung sounds (crackles or wheezes, evaluated for both hemithoraces), presence of abnormal tracheal sounds, coughing, signs of depressed attitude (subjective evidence of increased recumbence, lethargy, reluctance to rise), and nasal discharges were monitored, and results were recorded twice daily through 8 weeks (end of study). Thoracic ultrasonography was performed on the day of infection (prior to infection) and then weekly to identify evidence of peripheral pulmonary consolidation or abscess formation consistent with *R. equi* pneumonia. Foals were considered to have pneumonia if they demonstrated ≥3 of the following clinical signs: coughing at rest, depressed attitude (reluctance to rise, lethargic attitude, increased recumbency), rectal temperature of >39.4°C, respiratory rate of ≥60 breaths/min, or increased respiratory effort (manifested by abdominal lift and nostril flaring). Foals were diagnosed with *R. equi* pneumonia if they had ultrasonographic evidence of pulmonary abscessation or consolidation with a maximal diameter of ≥2.0 cm, positive culture of *R. equi* from T-TBA fluid and cytologic evidence of septic pneumonia from T-TBA fluid. The primary outcome was the proportion of foals diagnosed with *R. equi* pneumonia. Secondary outcomes included the duration of days meeting the case definition, the sum of total maximum diameters of ultrasound lesions (TMDs) observed at any single examination, and the sum of the TMDs for each examination over the study period. The TMD was determined by summing the maximum diameters of each lesion recorded in the 4th to the 17th intercostal spaces from each foal at every examination; the sum of the TMDs over time incorporates both the duration and severity of lesions. Foals diagnosed with *R. equi* pneumonia were treated with a combination of clarithromycin (7.5 mg/kg orally [p.o.] every 12 h [q12h]) and rifampin (7.5 mg/kg p.o. q12h) until both clinical signs and thoracic ultrasonography lesions had resolved. Foals also were treated as deemed necessary by attending veterinarians (A.I.B., N.D.C., and A.E.S.) with flunixin meglumine (0.6 to 1.1 mg/kg p.o. q12h to q24h) for inflammation and fever.

**Data analysis.** The effects of group (i.e., PNAG vaccination status of the mare), year of study, and year by group interaction on serum antibody and C´1q deposition activity of mares or foals were assessed using generalized linear modeling using the glm command with Gaussian link using R statistical software (44). Endpoint antibody activities were $log_{10}$ transformed to meet distributional assumptions of modeling. Correlations between antibody activities of PNAG antibody and C´1q deposition activity for mares were represented graphically, and the Pearson correlation coefficient was calculated along with a *P* value testing the hypothesis that the Pearson correlation coefficient was 0 using the ggpubr package in R. The proportion of foals that developed pneumonia was compared between the vaccinated and unvaccinated groups using a Fisher exact test using the fisher.test command in R. Significance for all analysis was set at a *P* value of <0.05.

## SUPPLEMENTAL MATERIAL

Supplemental material is available online only.
**SUPPLEMENTAL FILE 1**, PDF file, 0.3 MB.

## ACKNOWLEDGMENTS

This project was funded by the Morris Animal Foundation (D18EQ-015). Additional funding was provided by the Link Equine Research Endowment, and Noah D. Cohen is supported by the Patsy Link Chair in Equine Research.

We thank Angelica Allegro, Jocelyne Bray, Sophia Cortez, S. Garrett Wehmeyer, Alexis Gooch, and Shelby Zbranek for technical assistance.

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
