## [Reviewer comments · Microbiology Spectrum]

**Microbiology
Spectrum**

Serum Antibody Activity Against PNAG - But Not PNAG Vaccination Status - Is Associated with Protecting Newborn Foals Against Intrabronchial Infection with *Rhodococcus equi*

Noah Cohen, Susanne Kahn, Colette Cywes-Bentley, Sophia Ramirez-Cortez, Amanda Schuckert, Mariana Vinacur, Angela Bordin, and Gerald Pier

Corresponding Author(s): Noah Cohen, Texas A&M University

Review Timeline:

Submission Date:

June 21, 2021

Accepted:

July 7, 2021

Editor: Joanna Goldberg

Reviewer(s): The reviewers have opted to remain anonymous.

Transaction Report:

DOI: <https://doi.org/10.1128/Spectrum.00638-21>

July 7, 2021

Dr. Noah D Cohen
Texas A&M University
Large Animal Medicine & Surgery
College of Veterinary Medicine
Texas A&M University
College Station, TX 77843-4475

Re: Spectrum00638-21 (Serum Antibody Activity Against PNAG - But Not PNAG Vaccination Status - Is Associated with Protecting Newborn Foals Against Intra-bronchial Infection with *Rhodococcus equi*)

Dear Dr. Noah D Cohen:

Your manuscript has been accepted, and I am forwarding it to the ASM Journals Department for publication. You will be notified when your proofs are ready to be viewed.

Sincerely,

Joanna Goldberg
Editor, Microbiology Spectrum
